# Prevention of pancreatic acinar cell carcinoma by Roux-en-Y Gastric Bypass Surgery

Rui He[1,2], Yue Yin[1], Wenzhen Yin[1], Yin Li[1], Jing Zhao[1] & Weizhen Zhang[1,3]

Roux-en-Y Gastric Bypass Surgery (RYGB) prevents the occurrence of pancreatic cell acinar carcinoma (ACC) in male and female Ngn3-Tsc1−/− mice. Ngn3 directed Cre deletion of Tsc1 gene induced the development of pancreatic ACC. The transgenic mice with sham surgery demonstrated a cancer incidence of 96.7 ± 3.35% and survival rate of 67.0 ± 1.4% at the age of 300 days. Metastasis to liver and kidney was observed in 69.7 ± 9.7% and 44.3 ± 8.01% of these animals, respectively. All animals with RYGB performed at the age of 16 weeks survived free of pancreatic ACC up to the age of 300 days. RYGB significantly attenuated the activation of mTORC1 signaling and inhibition of tumor suppressor genes: p21, p27, and p53 in pancreatic ACC. Our studies demonstrate that bariatric surgery may limit the occurrence and growth of pancreatic ACC through the suppression of mTORC1 signaling in pancreas. RYGB shows promise for intervention of both metabolic dysfunction and organ cancer.

[1] Department of Physiology and Pathophysiology, School of Basic Medical Sciences, and Key Laboratory of Molecular Cardiovascular Science, Ministry of Education, Peking University Health Science Center, Beijing 100191, China. [2] Key Laboratory of Fertility Preservation and Maintenance of Ministry of Education, Key Laboratory of Reproduction and Genetic of Ningxia Hui Autonomous Region, School of Basic Medicine, and People's Hospital of Ningxia Hui Autonomous Region, Ningxia Medical University, Shengli Street No.1160, Yinchuan 750004, China. [3] Department of Surgery, University of Michigan Medical Center, Ann Arbor, MI 48109-0346, USA. These authors contributed equally: Rui He, Yue Yin. Correspondence and requests for materials should be addressed to W.Z. (email: weizhenzhang@bjmu.edu.cn)

Pancreatic cancer is one of the most leading causes of cancer deaths in the United States and many other developed countries[1]. Despite the comprehensive treatment options including surgery resection, radiotherapy and chemotherapy, the five-year survival rate remains steady at 3 to 5% in the past three decades[2]. Using the genetically engineered mouse models to recapitulate human pancreatic neoplasia, scientists have gradually uncovered the core signaling pathways and regulatory processes critical for the development of pancreatic cancers[3]. Mechanistic target of rapamycin (mTOR), a major regulator of cell growth, controls most anabolic and catabolic processes in response to nutrients and nutrient-induced signals[4,5]. mTOR activity is closely associated with the nutrient metabolism and growth of various tumors[4]. Two mTOR complexes have been identified: mTOR complex 1 (mTORC1) and mTOR complex 2 (mTORC2). mTORC1 contains mTOR, G protein-subunit-like protein, and Regulatory-associated protein of mechanistic target of rapamycin (Raptor). mTORC2 composes of mTOR and Rapamycin-insensitive companion of mTOR (Rictor)[6]. Recently, we have reported that activation of mTORC1 signaling by deletion of Tsc1 gene in cells expressing neurogenin 3 (Ngn3) results in the development of pancreatic acinar carcinoma[7]. Consistently, previous studies have demonstrated a close association of pancreatic neoplasia with hyper-activation of mTORC1 in both human beings and animals[8,9]. These observations suggest that mTORC1 is critical for the carcinogenesis of pancreatic cancers.

Contemporary treatment of pancreatic cancer is often ineffective because current understanding on the risk factors and etiology of pancreatic cancer from environmental and genetic perspective is limited[10]. In addition, pancreatic cancer is usually silent at the early stage. When diagnosed, most patients have already reached the advanced stage with metastasis to regional lymph nodes, and distant organs such as liver and kidney[10,11]. Roux-en-Y Gastric Bypass Surgery (RYGB) is increasingly applied to treat obesity and type 2 diabetes because of its significant and durable effects on weight and glycemic control[12,13]. Although RYGB has a profound metabolic benefit, its impact on tumor occurrence and development is rarely reported. A retrospective analysis of patients at a large tertiary bariatric surgery center has demonstrated a significantly lower organ cancer risk associated with greater weight loss after RYGB surgery[14]. Despite of this finding, there is currently no direct evidence supporting that RYGB surgery can effectively prevent the occurrence of cancers.

Using a mouse model of pancreatic acinar carcinoma[7]: Ngn3-Tsc1−/− transgenic mice in which tuberous sclerosis complex 1 (Tsc1) gene is deleted and mTORC1 signaling activated in neurogenin 3 (Ngn-3) positive cells, we here report that RYGB surgery blocks the spontaneous development of pancreatic cell acinar carcinomas (ACC).

## Results

### RYGB surgery prevents the occurrence of pancreatic cancer.
In the sham surgery group of Ngn3-Tsc1−/− male transgenic mice (Fig. 1a) in which Tsc1 gene is deleted and mTORC1 signaling activated (Fig. 1b) in Ngn-3 positive cells, spontaneous development of pancreatic ACC steadily increased over the time from 19.2 ± 5.9% to 45 ± 5.0% and 96.7 ± 3.5% at age of 150, 200, and 300 days, respectively (Fig. 1c). Similar incidence of pancreatic ACC was found for the sham surgery group of Ngn3-Tsc1−/− female transgenic mice (Fig. 1d). This incidence of pancreatic ACC in Ngn3-Tsc1−/− mice was consistent with our previous reports[7]. No pancreatic ACC was found in wild-type littermates at age up to 300 days. In order to examine the effect of RYGB surgery on the occurrence of pancreatic ACC, we performed RYGB surgery (Fig. 1e) in Ngn3-Tsc1−/− mice at the age of

16 weeks. This time point was chosen because no detectable cancer had occurred in mice before 16-weeks-old in these transgenic mice. Interestingly, RYGB surgery blocked the occurrence of pancreatic ACC in both Ngn3-Tsc1−/− male and female mice (Fig. 1c, d, f). Further, all Ngn3-Tsc1−/− male and female mice undergoing RYGB surgery survived at the age of 300 days, whereas male and female animals with sham surgery demonstrated a significantly lower survival rate of 68.4 ± 2.5% and 65.6 ± 3.6%, respectively (Fig. 1c, d). Pancreatic weight increased significantly in the Ngn3-Tsc1−/− mice with sham surgery, whereas it was comparable between Ngn3-Tsc1−/− mice with RYGB surgery and wild-type mice (WT) (Fig. 2a).

### RYGB surgery blocks the metastasis of pancreatic cancer.
We observed the metastasis of pancreatic ACC to distant organs such as liver and kidney in Ngn3-Tsc1−/− mice undergoing sham surgery. At age of 300 days, liver weight and kidney weight increased significantly in the Ngn3-Tsc1−/− mice with sham surgery, whereas no significant difference in liver and kidney weights between Ngn3-Tsc1−/− mice after RYGB surgery and wild-type mice (WT) was observed (Fig. 2a). In all, 69.7 ± 9.7% of the Ngn3-Tsc1−/− mice with sham surgery demonstrated liver metastasis (Fig. 2b), whereas 44.3 ± 8.0% showed metastasis to the kidney (Fig. 2d). Interestingly, we did not observe any metastasis to regional lymph nodes in the transgenic mice. The histological morphology of metastasized lesion was identical to the original cancer in the pancreas, with cellular and structure resemblance of pancreatic ACC (Fig. 2c, e). In animals with RYGB surgery performed at the age of 16 weeks, no cancer was observed in pancreas or liver and kidney up to the age of 300-days-old (Fig. 2).

### RYGB surgery alters body weight and glucose metabolism.
In our study, animals with RYGB surgery demonstrated only a slight reduction in body weight relative to WT animals (Fig. 3a). Although Ngn3-Tsc1−/− mice with sham surgery showed an obvious increase in post-operative body weight, net gain in body weight corrected with pancreatic cancer weight showed no significant difference relative to WT animals (Fig. 3b). These observations suggest that weight-independent mechanism may also contribute to the benefit effect of RYGB surgery on the control of pancreatic cancer.

Ngn3-Tsc1−/− mice demonstrated a significant improvement in glucose tolerance and insulin sensitivity before the occurrence of pancreatic ACC up to the age of 16 weeks[7]. Pancreatic cancer impaired the glucose metabolism as growth of pancreatic ACC was associated with a steady hyperinsulinemia and hyperglycemia in Ngn3-Tsc1−/− sham mice (Fig. 3c, d). RYGB significantly reversed the hyperinsulinemia and hyperglycemia in Ngn3-Tsc1−/− transgenic mice (Fig. 3c, d)

### Alteration in mTORC1 signaling and apoptotic proteins.
RYGB surgery significantly attenuated the deficiency of TSC1 on mTORC1 activation in either the pancreatic cancer tissue or its neighboring normal pancreatic tissues. Relative to the Ngn3-Tsc1−/− sham mice, Ngn3-Tsc1−/− RYGB mice showed a significant increase in levels of TSC1 in both cancer and normal pancreatic tissues (Fig. 4). Associated with this alteration is the reversal of mTORC1 activation in both normal and cancer pancreatic tissues. Phosphorylation of mTOR and S6 proteins was markedly attenuated in Ngn3-Tsc1−/− RYGB mice relative to Ngn3-Tsc1−/− sham animals (Fig. 4). Associated with the alteration in mTORC1 signaling, RYGB surgery significantly increased protein levels of apoptotic genes p21, p27, and p53

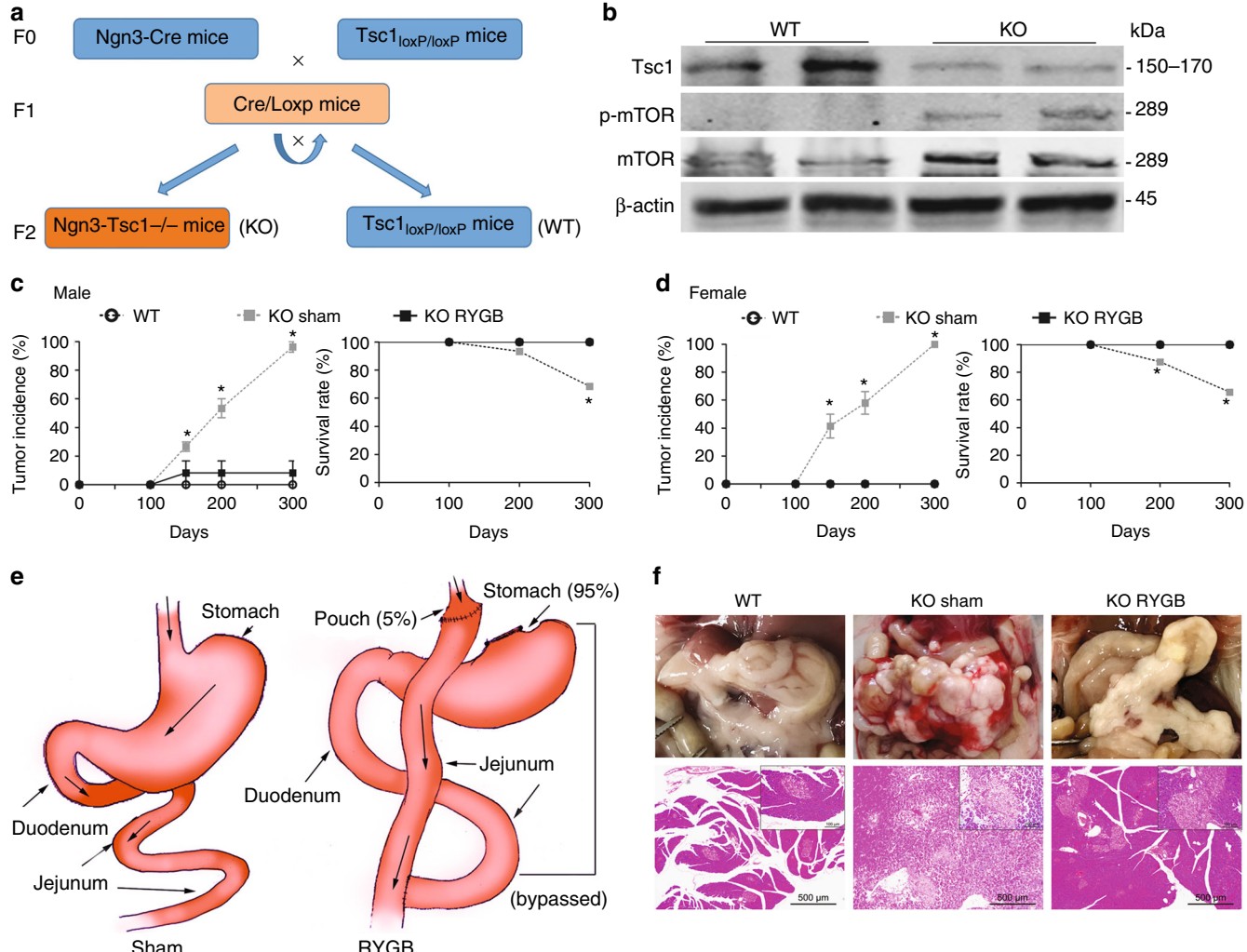

**Fig. 1** RYGB surgery effectively prevents the occurrence of pancreatic ACC. **a** Schematic diagram illustrating the generation of *Ngn3-Tsc1−/−* (KO) transgenic mice. **b** Tsc1, phospho-mTOR, and mTOR in pancreatic tissue of *Ngn3-Tsc1−/−* (KO) transgenic mice were examined by western blotting using specific antibodies. β-actin was used as loading control. *n* = 10, 6, 5 for WT sham, *Ngn3-Tsc1−/−*(KO) sham, *Ngn3-Tsc1−/−*(KO) RYGB, respectively. **c** Tumor incidence and survival rate of WT sham group, *Ngn3-Tsc1−/−*(KO) sham group, and *Ngn3-Tsc1−/−*(KO) RYGB group of male mice. Statistical differences were analyzed by log-rank (Mantel-Cox) test. *n* = 9, 5, 5 for WT sham, *Ngn3-Tsc1−/−*(KO) sham, *Ngn3-Tsc1−/−*(KO) RYGB respectively. *$P <$ 0.05 vs. *Ngn3-Tsc1−/−*(KO) RYGB. **d** Tumor incidence and survival rate of WT sham group, *Ngn3-Tsc1−/−*(KO) sham group, and *Ngn3-Tsc1−/−*(KO) RYGB group of female mice. Statistical differences were analyzed by log-rank (Mantel-Cox) test. *n* = 8, 5, 4 for WT sham, *Ngn3-Tsc1−/−*(KO) sham, *Ngn3-Tsc1−/−*(KO) RYGB, respectively. *$P <$ 0.05 vs. *Ngn3-Tsc1−/−*(KO) RYGB. **e** Schematic diagram illustrating sham (left) and RYGB (right) surgery. **f** Gross and histological morphology of normal pancreatic tissue and pancreatic ACC

in both normal and cancer pancreatic tissues relative to *Ngn3-Tsc1−/−* sham mice (Fig. 4).

**Effect of energy intake on the development of pancreatic ACC in *Ngn3-Tsc1−/−* mice.** *Ngn3-Tsc1−/−* mice fed with 45% high-fat diet (HFD, D12451; Research Diets Inc., New Brunswick, NJ) developed pancreatic ACC significantly earlier relative to those fed standard chow diet (NCD). Pancreatic ACC was detected in 66.7% of *Ngn3-Tsc1−/−* mice fed HFD at 84-days-old, whereas the transgenic mice fed NCD demonstrated no detectable cancer at this age (Fig. 5a). In addition, pancreatic weight increased significantly in *Ngn3-Tsc1−/−* mice fed HFD due to the occurrence of pancreatic ACC (Fig. 5b). Conversely, *Ngn3-Tsc1−/−* mice fed 70% calories of littermates demonstrated no detectable pancreatic ACC up to the age of 100 days.

## Discussion

Current therapy for pancreatic cancer is often ineffective. Our study demonstrates that RYGB surgery holds promise for the intervention of this malignancy. When performed before the development of pancreatic ACC at the age of 16 weeks, RYGB surgery is sufficient to block the spontaneous occurrence of pancreatic neoplasia in both *Ngn3-Tsc1−/−* male and female mice. No cancer was observed in pancreas or liver and kidney in *Ngn3-Tsc1−/−* transgenic mice up to the age of 300 days. However, *Ngn3-Tsc1−/−* mice with sham surgery demonstrate nearly 100% incidence of pancreatic ACC, low survival rate, metastasis to liver and kidney at a rate of 69.7 ± 9.7% and 44.3 ± 8.0%, respectively, at the age of 300 days. The high incidence in the liver metastasis rate may be due to the age of animals. In our previous report, animals were sacrificed and assessed for liver metastasis no older than 200 days. In the present study, the metastasis of pancreatic ACC to distant organs such as liver and

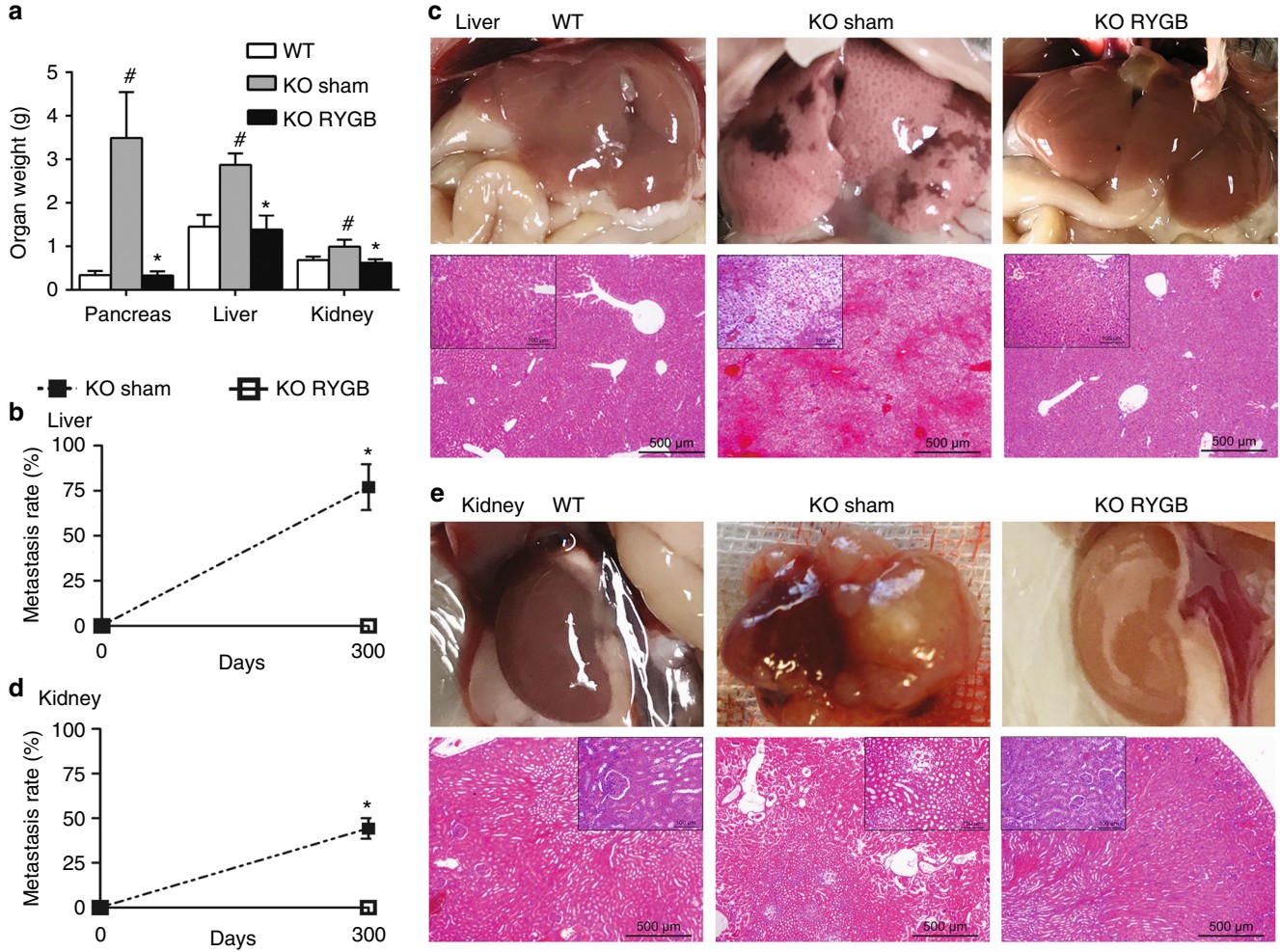

**Fig. 2** RYGB surgery effectively prevents the metastasis of pancreatic ACC. **a** Weight of pancreas, liver, and kidney. #*P* < 0.05 vs. WT. \**P* < 0.05 vs. *Ngn3-Tsc1−/−*(KO) sham. **b** Liver metastasis rate (left). **c** Liver gross and histological (H&E) morphology (right) in *Ngn3-Tsc1−/−*(KO) mice with sham or RYGB surgery. **d** Kidney metastasis rate (left). **e** Kidney gross and histological (H&E) morphology (right) in *Ngn3-Tsc1−/−*(KO) mice with sham or RYGB surgery. Values are mean ± SEM. *n* = 9, 5, 5 for WT sham, *Ngn3-Tsc1−/−*(KO) sham, *Ngn3-Tsc1−/−*(KO) RYGB, respectively, for male mice. *n* = 8, 5, 4 for WT sham, *Ngn3-Tsc1−/−*(KO) sham, *Ngn3-Tsc1−/−*(KO) RYGB, respectively, for female mice. Statistical differences were analyzed by one-way ANOVA followed by the Student's *t*-test. \**P* < 0.05 vs. *Ngn3-Tsc1−/−*(KO) RYGB

kidney in *Ngn3-Tsc1−/−* mice undergoing sham surgery or RYGB surgery was assessed at age of 300-days-old. In line with our observation, previous reports have shown that pancreatic cancer is usually silent at the early stage[7,10]. Most patients have already reached the advanced stage at the time of diagnosis. Over 85 percent of pancreatic cancers have extended beyond the pancreas to liver and peritoneum[10].

It is generally accepted that weight loss improves metabolic dysfunction, systemic inflammation, and adipokine profile. Thus, weight loss is intuitively proposed to result in physiologic functions, which would favor the reduction of cancer risk[15–18]. Consistent with this concept, a retrospective study at a large tertiary bariatric surgery center has recently revealed a positive relationship between the extent of weight loss after metabolic surgery and cancer risk reduction[19]. In our study, animals with RYGB surgery demonstrate only a slight reduction in body weight relative to WT animals (Fig. 3a). Although *Ngn3-Tsc1−/−* mice with sham surgery show an obvious increase in post-operative body weight, net gain in body weight corrected with pancreatic cancer weight demonstrates no significant difference relative to WT animals (Fig. 3b). These observations suggest that weight-

independent mechanism may also contribute to the benefit effect of RYGB surgery on the control of pancreatic cancer.

Hyperinsulinemia (either endogenous or exogenous) and hyperglycemia have also been proposed to affect the carcinogenesis[14], although its underlying mechanism remains unknown. Epidemiological studies have revealed a greater risk for organ cancers of the liver, pancreas, and endometrium imparted by type 2 diabetes[14,20]. For pancreatic cancer, it is difficult to interpret the causal nature of this association because that pancreatic cancer may result in dysfunction of glucose metabolism, a phenomenon called "reverse causality"[21]. The finding that positive association between diabetes and risk of pancreatic cancers is only restricted to diabetes that precedes the diagnosis of pancreatic cancer by at least 5 years does not support this reverse causation[22]. Consistent with this observation, our data indicate that hyperinsulinemia and hyperglycemia may not be the sole mechanism. First, *Ngn3-Tsc1−/−* mice demonstrate a significant improvement in glucose tolerance and insulin sensitivity before the occurrence of pancreatic ACC up to the age of 16 weeks[7]. Second, pancreatic cancer impairs glucose metabolism as growth of pancreatic ACC is associated with a steady presence of hyperinsulinemia and

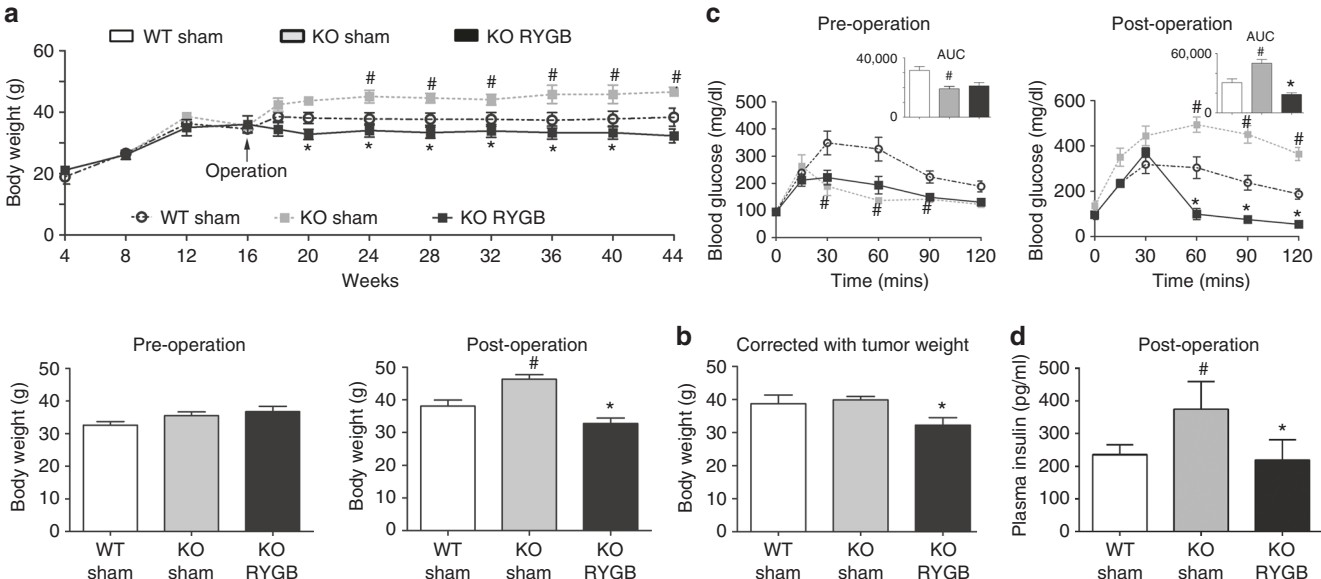

**Fig. 3** Body weight and glucose metabolism might before and after RYGB surgery. **a** RYGB surgery produced a moderate reduction in body weight. **b** Post-operative body weight corrected with pancreatic cancer weight. **c** Oral glucose tolerance test before operation (left) at the age of 16 weeks, and after operation at the age of 44 weeks (right). **d** Plasma levels of insulin after operation at the age of 44 weeks. Values are mean ± SEM. $n = 9, 5, 5$ for WT sham, *Ngn3-Tsc1−/−*(KO) sham, *Ngn3-Tsc1−/−*(KO) RYGB, respectively. Statistical differences were analyzed by one-way ANOVA followed by the Student's t-test. #$P < 0.05$ *Ngn3-Tsc1−/−*(KO) sham vs WT sham, *$P < 0.05$ *Ngn3-Tsc1−/−*(KO) RYGB vs. *Ngn3-Tsc1−/−*(KO) sham

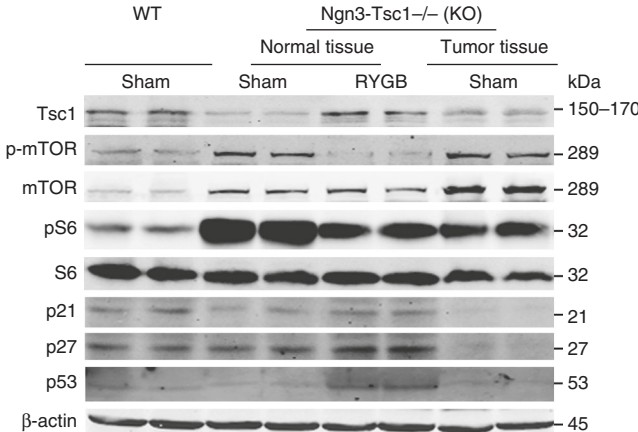

**Fig. 4** Alteration in mTOR signaling and apoptotic proteins. mTOR signaling molecules such as Tsc1, phospho-mTOR (p-mTOR), mTOR, phospho-S6 (pS6), S6, as well as apoptotic proteins including p21, p27, p53 in normal and tumor pancreatic tissues were examined by western blotting using specific antibodies. $n = 2$ for each condition, repeat two times

hyperglycemia in *Ngn3-Tsc1−/−* sham mice. All these observations indicate that mechanisms other than glucose and insulin may contribute to the reduction of pancreatic ACC induced by RYGB surgery.

As a downstream effector for many oncogenic pathways, mTORC1 plays an important role in the carcinogenesis. Hyper-activation of mTORC1 is associated with higher risk of human cancers[23]. Mutation of mTOR gene has been detected in several human cancers[24]. Deletion of ribosomal protein S6 renders the mice resistant to chemically or genetically induced pancreatic cancer precursor lesions[25]. Suppression of mTORC1 signaling inhibits the growth of pancreatic cancer cells[9]. Activation of mTORC1 signaling by deletion of *Tsc1* gene in pancreas leads to the occurrence of pancreatic ACC[7]. All these evidences indicate

that mTORC1 is critical for the occurrence of pancreatic ACC. Consistent with this concept, our data demonstrates that RYGB surgery significantly attenuates the deficiency of TSC1 on mTORC1 activation in either the pancreatic cancer tissue or its neighboring normal pancreatic tissues. Consistent with the alteration in mTORC1 signaling, RYGB surgery significantly increases protein levels of apoptotic genes p21, p27, and p53 in both normal and cancer pancreatic tissues relative to *Ngn3-Tsc1−/−* sham mice. Further, our previous studies have demonstrated that inhibition of mTORC1 signaling by rapamycin significantly attenuates the growth of the pancreatic neoplasm in *Ngn3-Tsc1−/−* mice[7]. All these data indicate that RYGB surgery may inhibit the occurrence and growth of pancreatic ACC by suppressing the mTORC1 signaling.

Over-nutrition has been demonstrated to increase mTORC1 signaling in a variety of tissues and cells[26]. Consistent with this concept, *Ngn3-Tsc1−/−* mice fed with 45% HFD develop pancreatic ACC significantly earlier relative to those fed standard chow diet (NCD). Conversely, *Ngn3-Tsc1−/−* mice fed 70% calories of littermates demonstrate no detectable pancreatic ACC up to the age of 100 days. All these observations further support that energy supply may alter the development of pancreatic cancer. Future studies should investigate whether mTORC1 mediates the effect of energy supply on the pancreatic cancer.

It is worth noting that mechanism other than mTORC1 may contribute to the prevention of pancreatic cancer by RYGB surgery. Indeed, use of glucagon-like peptide 1 (GLP1) analog has been reported to increase the occurrence of cancers in pancreas and colon, suggesting that increase of GLP1 is the culprit for the carcinogenesis[27]. However, studies from us (Supplementary Fig. 1) and others[28–30] have demonstrated that plasma GLP1 is significantly increased after RYGB surgery. If GLP1 contributes to the beneficial effect of RYGB on the occurrence of pancreatic ACC, one would have observed its decrease rather than increase after the surgery. Thus, it is unlikely that GLP1 underlies the prevention of pancreatic ACC by RYGB surgery.

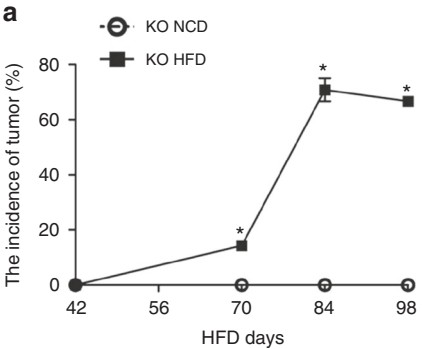
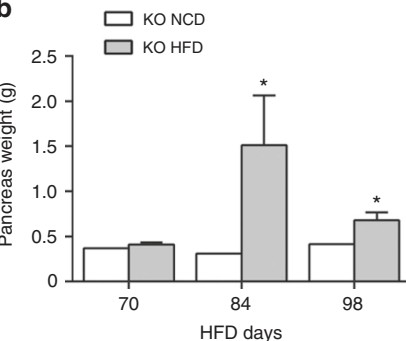

Fig. 5 Effect of HFD on the development of pancreatic ACC in *Ngn3-Tsc1−/−* mice. **a** Pancreatic ACC incidence. **b** Pancreatic weight *Ngn3-Tsc1−/−* (KO) mice at 6-weeks-old were fed with 45% HFD or NCD for 4, 6, and 8 weeks. Animals were sacrificed and pancreas assessed for ACC. $n = 8$ and 7 for 70-days-old KO mice fed HFD or NCD for 4 weeks, respectively; $n = 11$ and 7 for 84-days-old KO mice fed HFD or NCD for 6 weeks, respectively; $n = 7$ and 6 for 98-days-old KO mice fed HFD or NCD for 8 weeks, respectively. Values are mean ± SEM. *$P < 0.05$ *Ngn3-Tsc1−/−*(KO) HFD vs. *Ngn3-Tsc1−/−*(KO) NCD

In summary, the present study demonstrates that bariatric surgery limits the occurrence and growth of pancreatic ACC through the suppression of mTORC1 signaling in the pancreas. RYGB surgery may thus serve as the potential approach for intervention of both metabolic dysfunction and organ cancer.

## Methods

**Materials**. The antibodies used in this study are listed in Supplementary Table 1. Forty-five percent HFD (D12451) was from Research Diets Inc. (New Brunswick, NJ).

**Animals and animal care**. *Ngn3-Cre* mice that express the *Cre* recombinase gene under the control of the *Ngn3* gene promoter, as well as *Tsc1loxP/loxP* mice, in which exons 17 and 18 of the *Tsc1* gene are flanked by *loxP* sites by homologous recombination, were purchased from the Jackson Laboratory (Bar Harbor, ME). The *Ngn3-Tsc1−/−* transgenic mice were generated by breeding *Tsc1loxP/loxP* mice with *Ngn3-Cre* mice as described before[7]. Control experiments were performed using littermate *Tsc1loxP/loxP* animals. Deletion of the *Tsc1* gene was validated by the absence of *Tsc1* protein, and the increased phosphorylation of mTOR and S6, its downstream signaling molecules, in pancreas tissues. Mice were housed on a 12 h–12 h light–dark cycle. Normal chow and water were available ad libitum.

**Study approval**. The animals used in this study were reviewed and approved by the Animal Care and Use Committee of Peking University in accordance with the "Guide for the Care and Use of Laboratory Animals" published by the US National Institutes of Health (NIH publication no. 85–23, revised 1996).

**Genotyping**. For genotyping, genomic DNA was extracted from tail cuttings. Polymerase chain reactions were carried out for each animal to test for the presence of the *Tsc1* and/or the deletion of its exons 17 and 18 using *LoxP* primers, and *Cre* constructs using cre-specific primers, respectively[7].

**Surgical procedures**. For Roux-en-Y Gastric Bypass Surgery, mice were anesthetized with pentobarbital sodium (60 mg/kg body weight) and stomach exposed. Perigastric ligaments were ligated and cut to release the stomach. The left gastric vessel was separated bluntly from the cardia to make room for pouch operation without impairing the gastric blood supply. A titanium clip was applied to generate a small pouch size of about 5% of the total stomach volume. The stomach was transected right above the clip, leaving the left gastric vessels intact. Jejunum was transected about 2 cm distal to the ligament of Treitz. The small gastric pouch was then anastomosed to the cut end of the jejunum using 11-0 nylon suture in an uninterrupted suture fashion. For the jejuno-jejunostomy, a longitudinal slit was made on the anti-mesenteric side of the jejunum at 6 cm distal to the site of gastrojejunostomy, and the proximal end of the jejunum was joined in an end-to-side anastomosis using 11-0 nylon suture in an uninterrupted fashion. This resulted in a common limb consisting of the distal jejunum and the ileum of about 12 cm, a Roux limb about 5–6 cm and a biliopancreatic limb about 5–6 cm. In the abdominal wall, the muscular layer and skin were closed using interrupted 6-0 nylon suture and interrupted 5-0 nylon suture respectively.

For the sham operation, the stomach was released and the left gastric vessel separated from the cardia as described in RYGB procedures. The stomach, esophagus, and small intestine were exposed and the abdominal cavity closed with interrupted suture.

**Measurement of body weight and glucose metabolism**. Body weight was measured every week since 4-weeks-old.

Basal levels of glucose were measured using blood drawn from the tail vein 1 week before surgery and 1 week before killing. For oral glucose tolerance tests, mice were fasted for 16 h before gastric administration of glucose (3 g/kg body weight) by gavage. Blood was drawn from a cut at the tip of the tail at 0, 15, 30, 60, 90 and 120 min after glucose administration, and glucose concentrations were detected immediately. Area under the glucose curve (AUC) was calculated.

**Measurement of plasma insulin and GLP1**. Blood samples from mice were transcardially collected after anesthesia, immediately transferred to chilled polypropylene tubes containing EDTA-2Na (12.5 mg/ml) and aprotinin (1000 units/ml), and centrifuged at 1500 × g for 10 min at 4 °C. Plasma was separated and stored at –80 °C before use. Blood levels of insulin was measured using an enzyme linked immunosorbent assay (ELISA). Aprotinin was purchased from Amersham Biosciences (Pittsburgh, PA).

Active forms of GLP1 were assayed using the enzyme immunoassay kits according to the manufacturer's instructions (EGLP-35K; Billerica, MA).

Plasma insulin was measured using the insulin radioimmunoassay kit purchased from Linco Bioscience Institute (St. Charles, MO).

**Histological examination**. Pancreas, liver, and kidney tissues were harvested and fixed with neutral buffered formalin containing 4% formaldehyde for 6 h. Tissue sections were cut at 6 μm, adhered to charged slides, air dried for 5 min, and rehydrated with 0.01 M phosphate buffer saline (PBS). The sections were stained with hematoxylin and eosin for H&E staining.

**Western blotting**. Pancreas tissues extracts were immunoblotted with antibodies against Tsc1, phospho-mTOR, mTOR, phospho-S6, S6, p53, p21, p27, or β-actin. Specific reaction was detected using IRDye-conjugated second antibody and visualized using the Odyssey infrared imaging system (LI-COR Biosciences, Lincoln, NE). The antibodies are listed in Supplementary Table 1.

**Statistical analysis**. Data were presented as mean ± SEM and compared among groups using one-way analysis of variance (ANOVA). GraphPad Prism 6.0 software was used for statistical analysis. $P < 0.05$ denotes statistical significance.

## Data availability

The data supporting the findings of this study are available within the Article and Supplementary Files, or available from the authors upon reasonable request.

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

## Acknowledgements

This work was supported by grants from the National Key R&D Program of China (2017YFC0908900), the National Natural Science Foundation of China (81730020, 81330010, 81390354), and the National Institute of Health grant R01DK 112755-01A1.

## Author contributions

R.H. and Y.Y.: Data acquisition, analysis and interpretation, manuscript drafting; W.Z.: study design and supervision, manuscript revision, obtained funding; W.Y. Y.L., J.Z.: technical support.

## Additional information

**Competing interests:** The authors declare no competing interests.

