## [Peer Review File · Nature Communications]

Reviewers' Comments:

Reviewer #1:

Remarks to the Author:

In this manuscript, the authors describe the effects of Roux-en-Y gastric bypass surgery (RYGB) in the development of pancreatic cancer in mouse models. They use in the study the previously developed mouse that deletes *tsc1* expression in the pancreas, and thus hyperactivates the mTOR pathway. They show that RYGB in these mice results in decreased pancreatic cancer development and growth. These effects correlated with the normalization of mTOR activity. The experiments are well done and the results are significant and cannot be contested. Indeed the blockade of pancreatic cancer development is quite impressive. I have, however, significant concerns regarding the physiological relevance of this study.

1. RYGB is usually performed in morbid obese patients which, in addition to other pathologies, they have an increased risk to develop pancreatic cancer. In this regard, the approach that the authors suggest could benefit these subjects. The problem is that the authors use a model that does not represent this population. The physiological relevance of this model is, therefore, very low. An obese mouse model could better stand for this pathological risk. Do the *Tsc1*^{-/-} *Ngn3* mice have increased pancreatic cancer development when fed with high-fat diet?

2. In addition to this exciting observation, the authors do not provide with any mechanism underlying these effects. The explanation that mTOR activity is down is not sufficient. Some studies that analyze the metabolic impact of RYGB in obese subjects point to a significant impact on the GLP1 signaling. Strikingly, other studies have shown the implication of GLP1 in pancreatic cancer. This may be one clue. But other, more systematic studies, should be performed to elucidate the molecular mechanisms.

3. If RYGB surgery directly interferes with the mTOR pathway, then similar results would be observed when treating these mice with mTOR inhibitors.

4. Similarly, and more relevant, would the experiment analyzing the effects of caloric restriction or starvation in this mouse model. This will prove if the observed effects of RYGB surgery are mediated by reduced caloric intake, which blocks the mTOR pathway.

Reviewer #2:

Remarks to the Author:

This is good paper with important contribution. My concerns are only minor except that model of pancreas cancer used in non-standard and only previously reported by authors. the impact of this paper would be far greater if standard *kras*-induced model of PDA was used.

Minor concerns:

1. 1) *Ngn3*-*Tsc*^{-/-} mice have been reported to develop pancreatic acinar carcinomas (ACC). Please consider changing the nomenclature accordingly to ACC.

2. 2) In Fig. 1, please consider whether assessing pancreas weight as well as immunohistochemical quantification of pancreatic cancer could be more objective parameters.

3. 3) In Fig. 2, please report the results of liver and kidney weights.

4. 4) In Fig. 2, please clarify the high incidence of liver metastases: In a previous work, you have reported a metastasis rate of 5% (Ding et al., Neoplasia, 2014).

A Point-by-Point response

Reviewer #1 (Remarks to the Author):

In this manuscript, the authors describe the effects of Roux-en-Y gastric bypass surgery (RYGB) in the development of pancreatic cancer in mouse models. They use in the study the previously developed mouse that deletes *tsc1* expression in the pancreas, and thus hyperactivates the mTOR pathway. They show that RYGB in these mice results in decreased pancreatic cancer development and growth. These effects correlated with the normalization of mTOR activity. The experiments are well done and the results are significant and cannot be contested. Indeed, the blockade of pancreatic cancer development is quite impressive. I have, however, significant concerns regarding the physiological relevance of this study.

1. RYGB is usually performed in morbid obese patients which, in addition to other pathologies, they have an increased risk to develop pancreatic cancer. In this regard, the approach that the authors suggest could benefit these subjects. The problem is that the authors use a model that does not represent this population. The physiological relevance of this model is, therefore, very low. An obese mouse model could better stand for this pathological risk. Do the *Tsc1*^{-/-} *Ngn3* mice have increased pancreatic cancer development when fed with high-fat diet?

Reply: Additional experiments have been performed to examine the effect of high fat diet on the development of pancreatic cancer. We have found that pancreatic cancer in the transgenic mice fed HFD occurs significantly earlier relative to those fed NCD. This result is now included as figure 5 in line 167-173, page 5 in the result section. Our observation is consistent with previous reports demonstrating that morbid obese and diabetes patients have an increased risk to develop pancreatic cancer.

2. In addition to this exciting observation, the authors do not provide with any mechanism underlying these effects. The explanation that mTOR activity is down is not sufficient. Some studies that analyze the metabolic impact of RYGB in obese subjects' point to a significant impact on the GLP1 signaling. Strikingly, other studies have shown the implication of GLP1 in pancreatic cancer. This may be one clue. But other, more systematic studies, should be performed to elucidate the molecular mechanisms.

Reply: We agree with the reviewer that mechanism other than mTOR signaling may underlying the effect of RYGB surgery on the development of pancreatic cancers. However, following observations do not support that GLP1 mediates the effect of RYGB on the prevention of pancreatic cancers: (1) Increase of GLP1 has been reported as the culprit for the pancreatic cancer. (2) Studies from us (supplemental figure 1) and others¹⁸⁻²⁰ have demonstrated that RYGB significantly increases circulating GLP1. If GLP1 contributes to the beneficial effect of RYGB on the occurrence of pancreatic cancer, one would have observed its decrease rather than increase after the surgery. We have addressed this potential mechanism in lines 179-186, page 5 in the discussion section.

3. If RYGB surgery directly interferes with the mTOR pathway, then similar results would be observed when treating these mice with mTOR inhibitors.

Reply: Our previous studies have been demonstrated that inhibition of mTOR signaling by rapamycin significantly attenuates the growth of the pancreatic neoplasm². We have addressed this observation in lines 163-165, page 4 in the discussion section.

4. Similarly, and more relevant, would the experiment analyzing the effects of caloric restriction or starvation in this mouse model. This will prove if the observed effects of RYBG surgery are mediated by reduced caloric intake, which blocks the mTOR pathway.

Reply: We agree with the reviewer that caloric restriction would have a significant impact on the development of pancreatic cancer. Additional experiment has been performed to examine this potential. We have observed the animals fed 70% of calories for three months and detected no cancer at the present time. We have addressed this potential mechanism in lines 173-176, page 5 in the discussion section.

Reviewer #2 (Remarks to the Author):

This is good paper with important contribution. My concerns are only minor except that model of pancreas cancer used in non-standard and only previously reported by authors. the impact of this paper would be far greater if standard kras-induced model of PDA was used.

Reply: Thank for the encouragement. We do not possess the Kras PDA model but would be interested to pursue these experiments in the future.

Minor concerns:

1. 1) Ngn3-Tsc^{-/-} mice have been reported to develop pancreatic acinar carcinomas (ACC). Please consider changing the nomenclature accordingly to ACC.

Reply: We have made such change in the revision.

2. 2) In Fig. 1, please consider whether assessing pancreas weight as well as immunohistochemical quantification of pancreatic cancer could be more objective parameters.

Reply: We have now included the pancreatic weight in figure 2 (in line 98-100, page3) . The immunohistochemical features of this ACC model have been characterized in our previous report².

3. 3) In Fig. 2, please report the results of liver and kidney weights.

Reply: These information are now included in line 106-108, page3.

4. 4) In Fig. 2, please clarify the high incidence of liver metastases: In a previous work, you have reported a metastasis rate of 5% (Ding et al., Neoplasia, 2014).

Reply: The difference in the liver metastasis rate may be due to the age of animals. In our previous report, animals were sacrificed and assessed for liver metastasis no older than 200 days. In the present study, the metastasis of pancreatic acinar carcinoma to distant organs such as liver and kidney in Ngn3-Tsc1^{-/-} mice undergoing sham surgery or RYGB surgery was assessed at age of 300 days old.

Relative references

2. Ding, L. *et al.* Neurogenin 3-directed cre deletion of Tsc1 gene causes pancreatic acinar carcinoma. *Neoplasia* **16**, 909-917, (2014).
18. Hutch, C. R. & Sandoval, D. The Role of GLP-1 in the Metabolic Success of Bariatric Surgery. *Endocrinology* **158**, 4139-4151, (2017).
19. Zhai, H. *et al.* Takeda G Protein-Coupled Receptor 5-Mechanistic Target of Rapamycin Complex 1 Signaling Contributes to the Increment of Glucagon-Like Peptide-1 Production after Roux-en-Y Gastric Bypass. *EBioMedicine* **32**, 201-214, (2018).
20. Shah, M. *et al.* Contribution of endogenous glucagon-like peptide 1 to glucose metabolism after Roux-en-Y gastric bypass. *Diabetes* **63**, 483-493, (2014).

Reviewers' Comments:

Reviewer #1:

Remarks to the Author:

The authors have properly addressed the comments and suggestions of this reviewer. The manuscript has now enough quality for publication.

Reviewer #2:

Remarks to the Author:

The authors improved the manuscript from a mechanistic standpoint and have addressed most of my questions.

However, I have minor concerns left that warrant further experimentation prior to publishing:

1) In Fig. 2, please clarify the high incidence of liver metastases: You state that in your previous work the much lower incidence of ACC was due to earlier harvest (no later than 200 days). However, at 200 days the animals in this manuscript still seem to have a tumor incidence of about 50-60%. Could you further elaborate on the nature of the differences?

2) In new Fig. 5, please consider adding HFD in Tsc1^{-/-} Ngn3 vs HFD + RYGP in Tsc1^{-/-} Ngn3 mice.

A Point-by-Point response

Reviewers' comments:

Reviewer #1 (Remarks to the Author):

The authors have properly addressed the comments and suggestions of this reviewer. The manuscript has now enough quality for publication.

Reply: Thank you very much for the positive comments on our study.

Reviewer #2 (Remarks to the Author):

The authors improved the manuscript from a mechanistic standpoint and have addressed most of my questions.

However, I have minor concerns left that warrant further experimentation prior to publishing:

1) In Fig. 2, please clarify the high incidence of liver metastases: You state that in your previous work the much lower incidence of ACC was due to earlier harvest (no later than 200 days). However, at 200 days the animals in this manuscript still seem to have a tumor incidence of about 50-60%. Could you further elaborate on the nature of the differences?

Reply: We believe that the reviewer misunderstood the liver metastasis rate as the tumor incidence. In response to the reviewer's concern, we speculate that the difference in the liver metastasis rate (not the tumor incidence) may be due to the age of animals.

2) In new Fig. 5, please consider adding HFD in Tsc1^{-/-} Ngn3 vs HFD + RYGP in Tsc1^{-/-} Ngn3 mice.

Reply: We believe new Fig 5 already addresses the reviewer's concern on whether "the Tsc1^{-/-} Ngn3 mice have increased pancreatic cancer development when fed with high-fat diet" and its physiological relevance. I would therefore like to urge the reviewer to consider following challenges and allow us to publish our work without adding the data on HFD vs HFD+RYGB in Ngn3-Tsc1^{-/-} mice. (1) The experimental cycle for these studies is quite long because of

the nature of the ACC animal model and the low survival rate of RYGB in obese mice. (2) The surgeon who is capable of doing RYGB surgery has left the lab.